# Automatic Segmentation of Head and Neck Tumor: How Powerful Transformers Are?

**Ikboljon Sobirov**                                          IKBOLJON.SOBIROV@MBZUAI.AC.AE

**Otabek Nazarov**                                            OTABEK.NAZAROV@MBZUAI.AC.AE

**Hussain Alasmawi**                                          HUSSAIN.ALASMAWI@MBZUAI.AC.AE

**Mohammad Yaqub**                                            MOHAMMAD.YAQUB@MBZUAI.AC.AE

*Mohamed bin Zayed University of Artificial Intelligence, Abu Dhabi, UAE*

**Editors:** Under Review for MIDL 2022

## Abstract

Cancer is one of the leading causes of death worldwide, and head and neck (H&N) cancer is amongst the most prevalent types. Positron emission tomography and computed tomography are used to detect, segment and quantify the tumor region. Clinically, tumor segmentation is extensively time-consuming and prone to error. Machine learning, and deep learning in particular, can assist to automate this process, yielding results as accurate as the results of a clinician. In this paper, we investigate a vision transformer-based method to automatically delineate H&N tumor, and compare its results to leading convolutional neural network (CNN)-based models. We use multi-modal data from CT and PET scans to perform the segmentation task. We show that a solution with a transformer-based model has the potential to achieve comparable results to CNN-based ones. With cross validation, the model achieves a mean dice similarity coefficient (DSC) of 0.736, mean precision of 0.766 and mean recall of 0.766. This is only 0.021 less than the 2020 competition winning model (cross validated in-house) in terms of the DSC score. On the testing set, the model performs similarly, with DSC of 0.736, precision of 0.773, and recall of 0.760, which is only 0.023 lower in DSC than the 2020 competition winning model. This work shows that cancer segmentation via transformer-based models is a promising research area to further explore.

**Keywords:** cancer segmentation, head and neck tumor, CT, PET, multi-modal data, transformer-based segmentation, HECKTOR

## 1. Introduction

Head and neck (H&N) cancer is the eighth most common case of cancer mortality (O'rorke et al., 2012), and 686,328 people were diagnosed with H&N cancer worldwide in 2012 (Baijens et al., 2020). Clinically, positron emission tomography (PET) and computed tomography (CT) can be utilized to detect its presence. Doctors manually delineate the tumor region on 3D PET and CT scans and upon their analysis decide on a proper treatment (e.g. radiotherapy's dosage and location). Accurate detection of the tumor is crucial, however, since the data is volumetric, the process is highly time-consuming and challenging. Thus, automatic segmentation is a solution that is highly valuable for this task.

With the advances in artificial intelligence (AI) and deep learning (DL), automation of the tumor segmentation task has been studied with great interest. The primary reason of the popularity of DL in medical research field is that it can perform as good as a radiologist in most cases, and that it can save the time doctors spend to complete this task.

Even though H&N tumors is among the most frequent ones, it has an insufficient amount of studies that accurately segment out the tumor in the H&N area using DL techniques. Hence, Head and Neck Tumor Segmentation and Outcome Prediction in PET/CT images (HECKTOR) challenge (Andrearczyk et al., 2021) was proposed with automatic tumor segmentation being one of its tasks.

U-Net (Ronneberger et al., 2015), DeepLab (Chen et al., 2016) and their variations are generally used for segmentation task due to their accurate and fast performance. Their variations include features and techniques such as 3D convolution blocks (Çiçek et al., 2016), residual blocks (Zhang et al., 2017), multi-scale patches (Jiayun et al., 2018), ensembles (Feng et al., 2020). Most of the existing methods (Iantsen et al., 2021)(Yuan, 2020)(Ma and Yang, 2020) applied for H&N tumors segmentation in the HECKTOR challenge relied on U-Net variations. However, to the best of our knowledge, none of the work was applied with transformer-based models to explore the H&N tumor segmentation. Transformer-based models can be effective in this specific task because H&N tumor occurs only at specific regions, and transformers are superior at learning the context information of segmentation area and identifying potential tumors regions. Moreover, transformer-based models are relatively new and understudied compared to CNN-based models. Thus, exploring their performance further would potentially benefit other segmentation tasks. In this work, our contributions are as follows:

- Exploring a transformer-driven model in contrast to currently best performing CNN-based counterparts;
- Showing that the transformer-backed model is as powerful as CNN-based models;
- Showing that data augmentations are essential to the transformer-based model;
- Studying transformers in a CT/PET multimodal setting;
- Testing the validity of a newly proposed architecture in a different medical task.

## 2. Review of Related Work

The most common subject in research papers that use DL in medical imaging is the segmentation task (Litjens et al., 2017). Even though various DL architectures have been proposed, CNN-based U-Net (Ronneberger et al., 2015) and its variations have been consistently showing the best performance in this task in most cases. The vanilla U-Net model consists of an encoder to capture local contextual information, and a decoder to upsample back to the input image size, and skip connections between the two to restore spatial information. Interestingly, with an automatically configured architecture design and parameters, even the vanilla U-Net (Isensee et al., 2018) can show promising results, winning recent competitions as BraTS 2021 challenge (Menze et al., 2015).

### 2.1. Transformer-based Segmentation Models

Transformer-driven architectures are gaining more and more audience in the medical tasks. In the segmentation task, LeViT-UNet (Xu et al., 2021), TransBTS (Wang et al., 2021), CoTr (Xie et al., 2021), TransUNet (Chen et al., 2021), TransFuse (Zhang et al., 2021), UNet

TRansformer (UNETR) (Hatamizadeh et al., 2021) are some of the recent transformer-powered architectures that employ transformers mainly as a feature extractor, combining it with CNN either at the encoder or decoder paths. In particular, Wang et al. (Wang et al., 2021) effectively encodes local and global representations in depth as well as spatial dimensions to segment brain tumor. A 3D CNN encoder is used for spatial feature extraction, transformers following that for global feature encoding, and a 3D CNN decoder to upsample to the full resolution for segmentation. Another work by Zhu et al. (Zhu et al., 2021) integrates a region awareness into transformers to do breast tumor segmentation. Their extensive experimentation on ultrasound breast scans prove the model over CNN-based counterparts. UNETR, proposed in (Hatamizadeh et al., 2021), uses 12 layers of ViT transformer as an encoder that generates features at different layers and connects them to the decoder as skip connections, similar to the original U-Net. A CNN-based decoder upsamples the features to generate segmentation masks in the input size. The model is applied on brain tumor segmentation and abdominal multi-organ segmentation tasks, and achieves comparable results to other methods.

To the best of our knowledge, the study of transformers for H&N tumor segmentation is done for the first time. Furthermore, CT/PET multimodality has not yet been explored in these works. Lastly, our work is a validation study with a direct comparison of transformer-based and currently best performing CNN-based models.

## 2.2. Existing H&N Tumor Segmentation Models

Given the vast range of DL segmentation models, not much effort has been previously dedicated to the AI field to study automatic segmentation of H&N tumors. However, with the HECKTOR challenge, several papers attempted to design algorithms to automatically delineate tumor in H&N using PET and CT scans in a multi-modal approach.

Iantsen et al. (Iantsen et al., 2021) implements Squeeze-and-Excitation normalization (SE norm) layers on top of U-Net with residual blocks, achieving the highest dice similarity coefficient (DSC) of 0.759 in the 2020 HECKTOR challenge test set. The SE norm is similar to instance normalization (Ulyanov et al., 2016) but different in shift and scale values, which are treated as functions of input $X$ during inference. The SE norm is used in the decoder part after each convolutional block, and residual layers that contain the SE norm are used in the encoder part. They combine soft dice loss and focal loss for the training, and use ensembling for the final test set to achieve such a score.

An integration of U-Net and hybrid active contour is proposed by Ma and Yang (Ma and Yang, 2020) saving them the second place. They implement a model that combines CNN with a traditional machine learning technique, hybrid active contours, which aims to use the complementary information among CT images, PET images, and network probabilities to improve the segmentation results of the cases with high uncertainty. Their model shows similar performance to the best model in terms of DSC and precision metrics, but it does not provide a good recall value.

Yuan (Yuan, 2020) designs a dynamic scale attention network on top of U-Net to perform the segmentation. They argue that this helps enhance the utilization of feature maps coming from encoder to decoder. Their scale attention network (SA-Net) integrates different scale features by using a scale attention block for each decoder layer that is connected to all

extracted features (except the last encoder layer). They test their model and show that their model gets better results against the standard U-Net skip connection.

Nonetheless, all these approaches primarily focus on using CNN-based architectures to perform the H&N tumor segmentation. Transformers are understudied in the segmentation task, and in the H&N cancer task in particular.

## 3. Methods

### 3.1. Dataset

The HECKTOR 2021 challenge (Andrearczyk et al., 2021) provides a dataset of CT, FDG-PET, segmentation masks, bounding box information, and electronic health records (EHR). The data is collected from six medical centers. Total cases of 325 patients are provided with a split of 224 for training and 101 for testing by the organizers. Since we do not have access to the ground truth data of the test set, we split the training data into training and validation sets in a leave-one-center-out fashion. PET and CT scans are accompanied by a `CSV` file containing bounding boxes for tumor, which highlight the size of $144 \times 144 \times 144mm^3$ for each scan for consistency between CT and PET. For this task, CT, PET, segmentation masks, and the `CSV` file were used.

### 3.2. Image Pre-processing

The initial step was to utilize the bounding box information to crop the original scans down to the size of $144 \times 144 \times 144mm^3$, both in CT and PET. Given that this information was provided in the dataset, the full tumor appears within this cubic region, and the corresponding mapping of both modalities is accurate and without inconsistencies. The cropping vastly lowers the dimensions of the scans, highlights the tumor area, and removes redundant data in scans, assisting the models to learn more easily. Additionally, CT and PET image intensities were both normalized; CT images were initially clipped to (-1024, 1024), which were empirically chosen, and then normalized to (-1, 1); PET images were normalized using Z-score normalization. All scans were re-sampled to isotropic voxel spacing of $1.0mm$.

### 3.3. Data Augmentations

It is established that data augmentations contribute to the improvement in results, since it help the network see a variation of the existing data. With that in mind, several sets of augmentations were experimented with. Random rotation in (-45, 45) range, mirroring, zooming, gamma correction on PET, and elastic deformation were combined in various fashions and experimented with to investigate which combination yields the highest performance. Unlike other augmentations that were applied on both CT and PET, gamma correction was applied only on PET in 0.5-2 range due to the fact that PET scans are occasionally dark, and brightening and darkening them slightly can help the model understand PET scans in more details. Zooming was applied with a factor of 1.25, cropping the current size of $144 \times 144 \times 144mm^3$ to $115 \times 115 \times 115mm^3$. This was assumed to zoom into and highlight small tumor regions. Elastic deformation was experimented as well to make the model more robust as claimed in the original U-Net paper (Ronneberger et al., 2015).

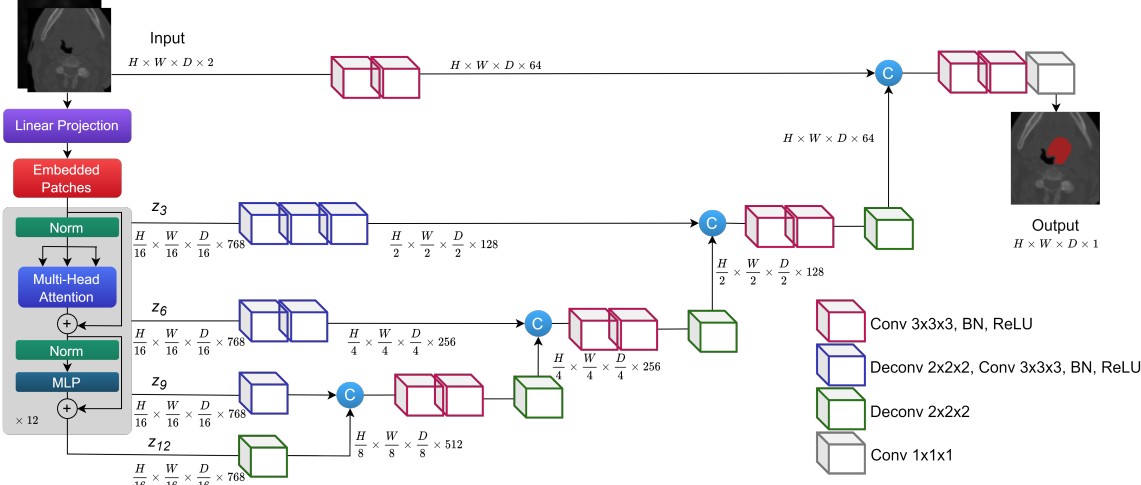

Figure 1: The figure shows the UNETR model architecture. CT and PET scans are inputted to the network as two channels. Output is a single channel mask. (Note that the output mask is superimposed on CT for better visualization). Inspired by (Hatamizadeh et al., 2021).

### 3.4. Transformer-based Method

Since their success in NLP, transformers have started to gain a lot of interest in the computer vision community. Inspired by UNETR (Hatamizadeh et al., 2021), we developed a transformer-based model in the task of H&N tumor segmentation to compare it to solely CNN-empowered models. UNETR is a transformer-driven encoder and CNN-based decoder model. To further explain it, a 3D input image gets split into several flattened uniform non-overlapping patches and embedded with a linear layer before going into the transformer block. The transformer layers are the same as the original ViT architecture (Dosovitskiy et al., 2021), with input normalization, multi-head attention, and multi-layer perceptron sub-layers. The transformers produce outputs at different layers (0, 3, 6, 9, and 12), 0 being the original input image and 12 being the last layer output, as skip connections to the decoder path. Before getting concatenated to the decoder path, these outputs undergo 3D upsampling and 3D convolution blocks to get to the desired sizes. Lastly, the final output is passed through a $1 \times 1 \times 1$ convolutional block followed by a softmax activation function to reach voxel-level segmentation. The architecture is depicted in Figure 1.

In our experiments, UNETR has `ViT-B16` model as a backbone with 12 layers, an embedding size of 768, a patch resolution of $16 \times 16 \times 16$. All the UNETR models were trained with the batch size of 8, utilizing an `AdamW` optimizer with a learning rate of `1e-3` for 800 epochs. The reason as to why the model is trained for 800 epochs was only to explore it in depth. A combination of soft dice loss and focal loss is used for training. Input and output channels were adapted for our task, with two input channels for CT and PET and one output channel for the segmentation mask.

Table 1: The table shows results of using different data augmentations with our UN-ETR model. Model performance is presented when using different data augmentations. NA=No Augmentation, MR=Mirroring, RT=Rotation, ZM=Zooming, GC=Gamma Correction, ED=Elastic Deformation.

| | NA | MR,RT | MR,RT,ZM | MR,RT,GC | MR,RT,ED | MR,RT,ZM,GC | MR,RT,ZM,GC,ED | MR,RT,GC,ED |
|---|---|---|---|---|---|---|---|---|
| DSC | 0.741 | 0.791 | 0.777 | 0.788 | 0.788 | 0.775 | 0.784 | **0.794** |
| Precision | 0.726 | 0.775 | 0.742 | **0.778** | 0.768 | 0.767 | 0.765 | 0.761 |
| Recall | 0.805 | 0.834 | 0.850 | 0.829 | 0.845 | 0.822 | 0.850 | **0.861** |

## 4. Results and Discussion

**Metrics.** As metrics of comparison, we report the dice similarity coefficient (DSC), precision and recall. The main reason behind this is that the challenge organizers assess the quality of the model using these metrics. On top of that, they can be used to compare our work to other previous work done in a similar domain.

**Augmentation Results.** The transformer-driven model was trained with several sets of data augmentations as well as with no augmentation to validate the importance of augmentations. For this group of experiments, we are using only one split of data (169 for training and 55 for validation) since our goal is to identify the right set of augmentations. Table 1 lists down the results of utilizing these data augmentations. With no augmentation, the model could achieve the DSC score of 0.741, precision of 0.726 and recall of 0.805. With all the augmentations, the results showed improvement in all three metrics, proving how important the augmentation process is for our model. Zooming with a factor of 1.25 did not show much improvement. It is hypothesized that zooming helps with information preservation when downsampling using CNNs, but UNETR downsamples using a ViT backbone, and zooming did not show effectiveness in case of transformers-based downsampling. The combination of mirroring, rotation, gamma correction on PET scans, and elastic deformation yielded the highest DSC and recall of 0.794 and 0.861 respectively, and precision of 0.761 for this split. Mirroring and rotation contributed the most to the score improvement because the head and neck structure of human is symmetric, and they would expose the model to a wider range of realistic samples, making the model more robust to unseen data. Lastly, gamma correction and elastic deformation, when combined, had a slight contribution to the score improvement because they made the model more exposed to varying PET intensities and shapes of tumor.

**Cross Validation Results.** The UNETR model was cross validated in a leave-one-center-out fashion. Since the training data comes from five different centers (sixth center is kept for testing, which we are not using), this approach of cross validation is preferred over, for example, $k$-fold cross validation. The reason of this choice lies at the fact that the model is supposed to be robust to new data, and training it on four random centers and validating on a different one tests this hypothesis. Another reason is that the total number of scans is different in each center of five. This forces the model to learn with various numbers of training and validation data, imitating a real-world scenario. UNETR is trained using the highest achieving data augmentations: a set of mirroring, rotation, gamma correction

Table 2: Dice, precision and recall of different models on the validation set are shown on the left. The dice, precision and recall of the UNETR model on the testing set are shown on the right. Mean and standard deviation are reported for the three models (left) which were trained and cross validated from scratch to provide a fair comparison.

| | Validation Set (5-Fold Cross Validation) | | | Testing Set |
|---|---|---|---|---|
| | SE-based U-Net | nnU-Net | UNETR | UNETR |
| DSC | $\mathbf{0.757}_{\pm 0.048}$ | $0.748_{\pm 0.061}$ | $0.736_{\pm 0.043}$ | 0.736 |
| Precision | $\mathbf{0.748}_{\pm 0.060}$ | $0.768_{\pm 0.095}$ | $0.766_{\pm 0.022}$ | 0.773 |
| Recall | $0.784_{\pm 0.044}$ | $\mathbf{0.788}_{\pm 0.031}$ | $0.766_{\pm 0.058}$ | 0.760 |

and elastic deformation. The model performed slightly different in metrics for each split, reaching a mean DSC of 0.736 ($\pm 0.043$), precision of 0.766 ($\pm 0.022$), and recall of 0.766 ($\pm 0.058$) as reported in Table 2.

**Comparison of Models.** The transformer-based model results are compared to CNN-driven models. In particular, two models are selected and trained as the original works: SE-based U-Net (Iantsen et al., 2021) that won the 2020 competition, and nnU-Net (Isensee et al., 2018) that claims that the model is automatically configured to the task and generates the highest results U-Net can ever produce. The SE-based U-Net, nnU-Net, and UNETR were all trained from scratch and cross validated in a leave-one-center-out fashion. UNETR achieves a mean DSC of 0.736 ($\pm 0.043$ standard deviation) that is only 0.012 and 0.021 lower than the powerful nnU-Net and SE-based U-Net respectively, proving the capability of the transformer-based model when trained from scratch.

**Testing Results.** The organizers of the HECKTOR committee have kindly checked the UNETR predictions on the testing set. Therefore, we were able to obtain the results on the testing set. The metrics are highly similar to the cross validation results, with a DSC, precision and recall of 0.736, 0.773, and 0.760 (compared to 0.736, 0.766, 0.766 in the validation set respectively).

**Qualitative Results.** Further qualitative results were conducted to understand the models' outputs. Prediction masks from each of the three models were compared to each other as well as to the ground truth. It is illustrated in Figure 2 that all the three models produce very similar segmentation masks with very minor differences, and that is the case for most samples. First line of images show a sample where all the three models performed extremely well, whereas the lower example shows their incapabilities. It is noteworthy that all the models are heavily dependent on both CT and PET. CT provides sufficient structural information so that the models can locate the tumor region with respect to the background body structure, and PET provides clarity in intensity differentiation for the model to accurately pinpoint the tumor location as is distunguishable in Figure 2.

**More Qualitative Results.** Figure 5 in the Appendix shows examples with which the models struggled to segment. Such inaccuracies occur with only a certain set of scans. To further investigate as to why all the models are struggling to segment the tumor regions in

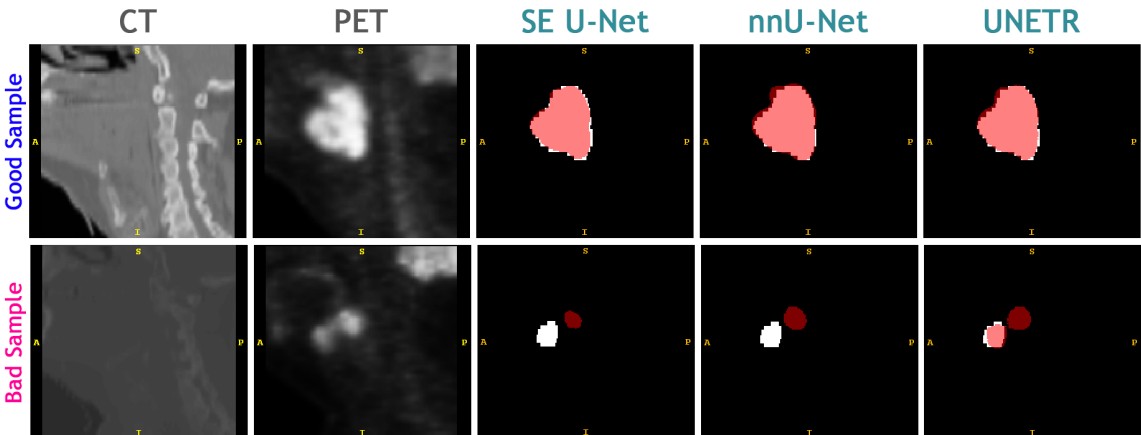

Figure 2: Segmentation examples where the models performed well and poorly. White and red represent the ground truth mask and model's prediction mask respectively.

these specific cases, the scans with the lowest DSC were extracted and examined. Primarily, three major commonalities are the reasons for the struggle of the models, as exemplified in Figure 4 in the Appendix. First, when the PET scan does not have a well-outlined intensity value at the region of the tumor, the models often produce a faulty output, segmenting another region with high PET intensity values. Second, the tumors are fairly smaller in size in these scans than other scans with high scores. When detecting tiny regions with a larger background abundant in data, the models commonly find it hard to figure out where to focus on. Finally, most of these scans suffer from artifacts, mainly streak, in CT that introduce irregularities in data, thereby causing the model to missegment. The streak artifacts are present near the tooth area, and are presumed to be caused by dental implants.

## 5. Conclusion

Automation of H&N tumor segmentation is a crucial task that should be studied in details. In this work, we studied a transformer-based model to tackle this problem and investigate its performance with respect to two common CNN models. We showed that transformers can come close to CNNs, reaching similar results. The transformer-driven model achieved a mean DSC of 0.736 ($\pm$0.043), precision of 0.766 ($\pm$0.022), and recall of 0.766 ($\pm$0.058) when it is trained from scratch. On the HECKTOR testing set, the model showed similar results (DSC of 0.736, precision of 0.773, and recall of 0.760). Reported results suggest that the utilized vision transformer network is slightly less accurate than well-mature CNN-based architectures. Although this can be considered a limitation, we believe that CNNs have gone through several improvements over the last few years while transformers are yet to be investigated in details. Furthermore, since self-supervised pretraining has been a key for transformers' success in the NLP domain, we hypothesize that self-supervised based pretraining should be further explored in case of transformer-based models in image segmentation.

## Acknowledgments

We thank the organizers of the HECKTOR challenge, Vincent Andrearczyk in particular, for their help with providing the results for the testing set. We also thank our colleagues Numan Saeed and Hashmat Shadab Malik for their fruitful discussions for the improvement of the paper.

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

**Appendix**

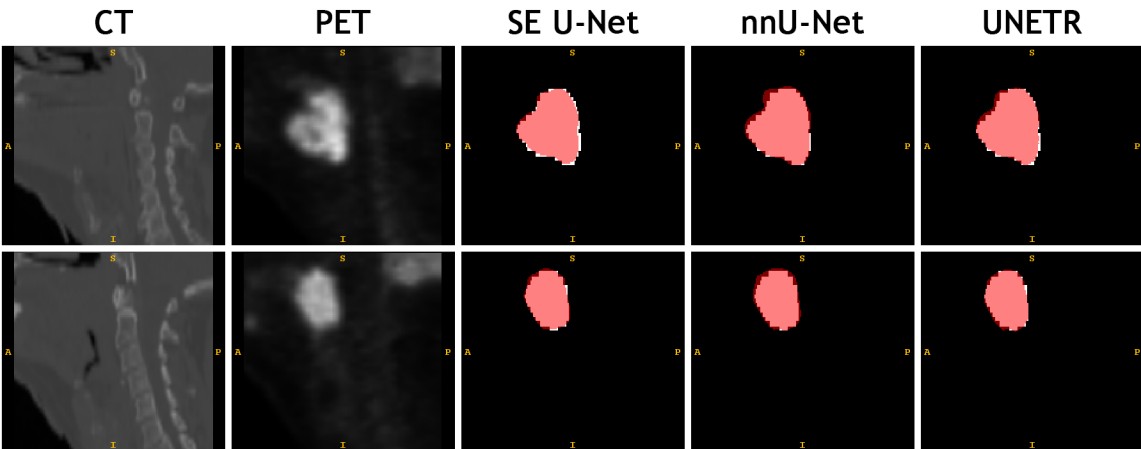

Figure 3: The figure shows segmentation examples where the models performed well. White represents the ground truth mask and red represents the model's prediction mask. All the models mostly produce these kinds of segmentation results. SE-based U-Net and UNETR segments the tumors very accurately, while nnU-Net over-segments by a small extent.

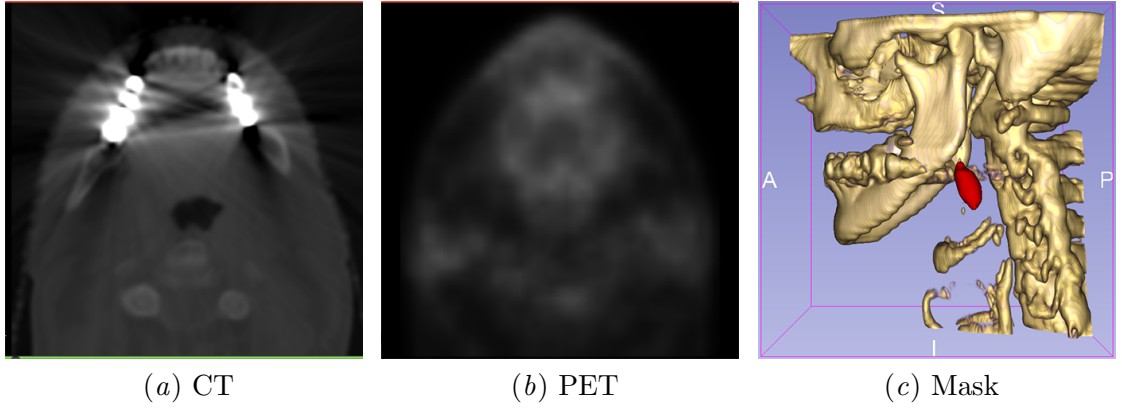

$(a)$ CT $\qquad$ $(b)$ PET $\qquad$ $(c)$ Mask

Figure 4: The figure depicts one sample with which the models struggled to segment; $(a)$ shows a CT slice with artifacts, $(b)$ shows an unclear PET slice, and $(c)$ shows a small sized mask (in red) superimposed on the CT bone structure. Note that this is a single scan, containing all the three issues.

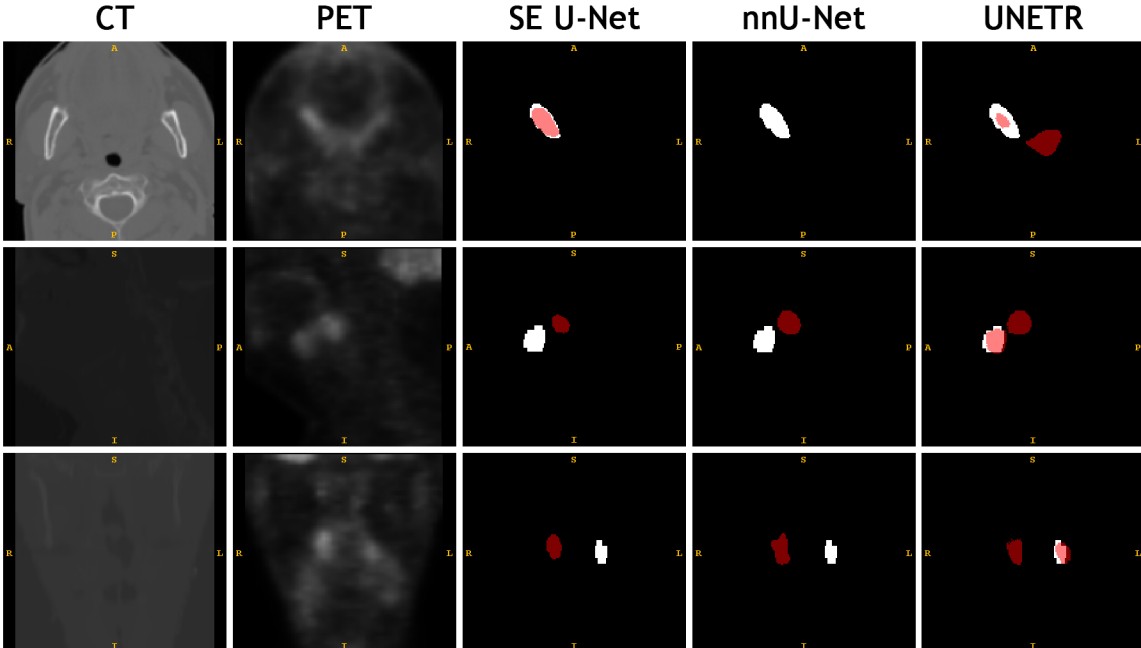

Figure 5: The figure shows segmentation examples where the models did not perform well. White represents the ground truth mask and red represents the model's prediction mask. The models fail to accurately segment the tumor on account of the unclarity in PET and CT scans and the small size of tumors. UNETR model can partially locate the tumor region in the samples, whereas the other two models fail to do that. SE-based U-Net occasionally shows better output than UNETR.

