# OpenReview forum: "Automatic Segmentation of Head and Neck Tumor: How Powerful Transformers Are?"
_MIDL.io/2022/Conference — MIDL 2022_

### Official Review · Reviewer_Mmhp · 2022-01-11

**Confidence:** 5
**Preliminary Rating:** 3
**Recommendation:** Poster

**Summary:**

This paper explores the use of transformers for the segmentation of H&N tumors in PET/CT images. It is relatively well written and easy to follow except for some typos and grammar mistakes to fix. The paper is interesting as a short study of transformers for this task, but the results are lower than the SoA and not comparable to the 2020 and 2021 HECKTOR challenge results (cross-validation on the training set instead of test results). The best cross-val models should be ensembled and submitted to the challenge to report the results on the test set of HECKTOR 2021 for a fair comparison with all other methods.
It is good, however, that the authors compared the SE-based UNet and the nnUnet on the same cross-validation setting, but not sufficient I think.


**Strengths:**

The paper proposes an interesting (first) use of transformers for this task of H&N tumor segmentation.
The paper is well written and structured. The evaluation with cross-validation, DSC, precision and recall, and visual examples is good (although test results should be reported).


**Weaknesses:**

- No evaluation on the test set which does not allow for a comparison with all other participants of the HECKTOR challenge.
- Results are lower than the SoA and the claim of performance on par is not supported by the results.


**Deanonymize Review:**

no

**Detailed Comments:**

- The authors should not claim par results with the SoA, it is not the case. I agree with the fact that CNNs have received a lot more developments and finetuning in the last decades than ViTs, although ViT models inherit a good part of it.
- In the abstract, the results are not comparable with HECKTOR 2020 winning results, it is misleading.
- In 3.2, describe better the normalization. Is the CT clipped? Is it Z-score normalization?
- Rotation in the range 0-45° seems too large since the orientation is fixed at test time.
- Specify mm^3 when mentioning the image sizes (e.g. 144x144x144mm^3, or voxels at 1mm^3 spacing)
- Add standard deviations to tables since the DSC is always an average across cases in the reported results.
- In 4.3, the fact that a baseline gets lower results does not show that your method (UNETR) is on par with the top performance.
- The figures representing the PET/CT images and volumes could be improved by e.g. setting a better CT range and maybe overlapping the predicted and GT volumes with the images.

**Final Rating After The Rebuttal:**

4: Weak Accept

**Justification Of The Final Rating:**

The authors answered all my comments. The quality has been improved. The results are lower than the best CNN models on this dataset and the insights are limited, yet sufficient for a conference paper I think. It is an interesting application of transformers, with good comparison with SoA models (final comparison on the test set should be available in the camera-ready paper), ablation study and good qualitative and quantitative results.

**Paper Type:**

validation/application paper

**Questions To Address In The Rebuttal:**

- The authors should report their results on the HECKTOR 2021 test set as suggested above to allow for a comparison with all participants/methods of the challenge
- Address also all comments above to improve the clarity of the paper.


**Special Issue:**

no

---

### Official Review · Reviewer_AN7B · 2022-01-20

**Confidence:** 3
**Preliminary Rating:** 3
**Recommendation:** Poster

**Summary:**

The paper proposes a validation study of U-net based Transformer model with application in Hean & Neck Tumour segmentation. The paper validates the usefulness of transformer U-Net by experimenting on a publicly available challenge dataset. The results show that transformer-based networks can give similar results to CNN.  Overall, paper is a good validation study with clear focus and good experimental process.

**Strengths:**

* The paper is clearly written with a clear focus as a validation study.
* Experimental setup is done on a publicly available challenge dataset.
* A good ablation study on performance difference of model when different data augmentation is applied. This is indeed necessary as the performance of the Transformer based model is notoriously dependent on the data augmentation.
* Performance of Transformer based model is clearly compared against the most popular nn-Unet based mode. In the discussion, it is clearly mentioned that the Transformer based model is not giving better performance but still it is an active area of research. This is good as there is no over-the-top claims.

**Weaknesses:**

* Although the literature review is commendable, it focuses too much time and space on the H&N tumor segmentation literature review instead of the Transformer based literature review. It would have been more beneficial for the authors to focus on transformer application in the medical image segmentation task.
* Clarity regarding what is already done in Transformer based deep learning architecture for medical image segmentation task, with what is missing from these papers, and how those things are addressed in this paper would put the paper in clear light and allow for a better understanding of the usefulness of the work.
* Discussion regarding why one type of data augmentation might be more useful compared to other is missing.
* Qualitative results are given in the appendix while the discussion regarding the same is given in the main paper. Maybe it would be helpful if the authors move at least one of the qualitative results to the main text.

**Deanonymize Review:**

no

**Detailed Comments:**

* There is no statistical significance analysis performed. Authors should either provide this or at least the standard deviation for each of the reported results.
* Did the authors use similar data augmentation techniques for all three networks in Table-1?
* Results for 2D and 3D U-net-based network is directly taken from a previously published work. Did the author use the same validation method (leave-one-center-out)?
* The results using Transformer based neural network don't seem to improve. Can authors comment on where Transformer based models might be more useful compared to CNN models?


**Final Rating After The Rebuttal:**

3: Borderline

**Justification Of The Final Rating:**

The authors have made sufficient changes to the paper. The paper is indeed a good validation study. I am still not that confident about the usefulness of the paper as it is neither applying the transformer-based method to a different and challenging task nor provides better insights into what is different compared to a lot of already studied transformer-based papers on various medical imaging tasks.

**Paper Type:**

validation/application paper

**Questions To Address In The Rebuttal:**

Mainly point 2,3, and 4 from the weakness section and Point 1 of the detailed comment section. Overall the paper is a good validation paper with a clear focus, but clarifying these points will make it a better one,

**Special Issue:**

no

---

### Official Review · Reviewer_egSq · 2022-01-24

**Confidence:** 3
**Preliminary Rating:** 4
**Recommendation:** Poster

**Summary:**

This work presents the application of transformer-based UNet architecture (UNet with a transform as the encoder) for segmentation of head and neck tumor. The experiments are done using data from the HECKTOR challenge. An extensive set of experiments comparing different data augmentation techniques as well as comparison to SE-Unet and nn-UNet is shown.

**Strengths:**

- The question about the performance of transformer models for medical imaging data to be quite relevant at the moment.
- Relatively well written paper and easy to follow.
- Comparison (qualitative and qualitative) to the winning method of the HECKTOR challenge as well as nn-Unet.


**Weaknesses:**

- The paper uses data from the HECKTOR challenge. The challenge seems to be closed for new submissions at the moments so the authors use only the training set and make their own experiments. This is clearly states in the text but I feel that it should be also stressed in the caption of the results table as to discourage direct comparison to the results of the challenge (which would not be fair).
- In Tabel 3, the results of the "vanilla" UNets are taken from the challenge overview paper. Again, this is on another dataset and I feel that It makes things confusing. It would be best to remove them.


**Deanonymize Review:**

no

**Detailed Comments:**

- It seems that like the average evaluation measures in Table 2 should be the same as the results in Table 3. Have the authors here made a typo - incorrectly copying 0.784 to 0.748 (or the other way round)?

**Final Rating After The Rebuttal:**

4: Weak Accept

**Justification Of The Final Rating:**

The authors have addressed my comments and the results in the manuscript are more clear and transparent (particularly w.r.t. the comparison with the challenge dataset). However, given the overall quality of the work I think my original rating still stands for this paper).

**Paper Type:**

validation/application paper

**Questions To Address In The Rebuttal:**

Mainly, the issue with using the training data needs to be even more transparent. The confusion (or possible type) with the results in Table 2 and 3 can also be addressed. I wonder of Table 2 is even needed, it would be probably better to show the average + SD instead of the individual forms and show everything in one table.

**Special Issue:**

no

---

### Meta-Review · Area_Chair_26CP · 2022-02-18

**Recommendation:** Accept (Poster)
**Confidence:** 5

**Metareview:**

This paper investigates the relevance of transformer networks for segmentation in the context of Head and Neck tumors. All reviewers recognized that the paper is relatively well written but also the need to report the performance on HECKTOR 2021's test set. This is important since the transformers seem not to outperform classical CNNs. This is an important (negative) result to report but would be more impactful if reported on the test set. I recommend the acceptance of the paper but highly suggest the inclusion of test performance in the camera ready and adaptation of the conclusions if needed.

---

### Decision · Program_Chairs · 2022-02-28

Accept